# Experimental Investigation on Effect of Temperature on FDM 3D Printing Polymers: ABS, PETG, and PLA

Ryan Mendenhall and Babak Eslami *

Mechanical Engineering Department, Widener University, One University Place, Chester, PA 19013, USA; rwmendenhall@widener.edu
* Correspondence: beslami@widener.edu

**Abstract:** Four-dimensional printing is a process in which a 3D-printed object is intentionally transformed in response to an external stimulus such as temperature, which is useful when the final geometry of a 3D-printed part is not easily manufacturable. One method to demonstrate this is to print a part made of thin strips of material on a sheet of paper, heat the part, and allow it to cool. This causes the part to curl due to the difference in the thermal expansion coefficients of the paper and plastic. In an attempt to quantify the effect of different temperatures on various materials, samples of three common 3D printing filaments, acrylonitrile butadiene styrene (ABS), polyethylene terephthalate glycol (PETG), and polylactic acid (PLA), were heated at different temperatures (85 °C, 105 °C, and 125 °C) for intervals of 15 min and then allowed to cool until curling stopped. This heating and cooling cycle was repeated three times for each sample to determine if repeated heating and cooling influenced the curling. Each sample was filmed as it was cooling, which allowed the radius of curvature to be measured by tracking the uppermost point of the part, knowing the arc length, and calibrating the video based on a known linear length. After three cycles, all three materials showed a decrease in the radius of curvature (tighter curl) as heating temperature increased, with PLA showing the trend much more predominantly than ABS and PETG. Furthermore, for PETG and PLA, the radius of curvature decreased with each cycle at all temperatures, with the decrease being more significant from cycle 1 to 2 than cycle 2 to 3. Conversely, ABS only shared this trend at 125 °C. The findings of this work can provide guidelines to users on the temperature dosage for the mass manufacturing of complex geometries such as packaging, self-assembly robots, and drug delivery applications.

**Keywords:** 4D printing; polymer; PLA; ABS; additive manufacturing; FDM 3D printing

## 1. Introduction

Since the invention of the first technology related to additive manufacturing, known as 3D printing, the opportunities in machine design, automation, bioengineering, etc., have expanded [1–3]. The mostly common 3D printing technique known as fused deposition modeling (FDM) is based on depositing thermoplastic polymers such as polylactic acid (PLA) and acrylonitrile butadiene styrene (ABS) layer by layer according to a given computer-aided design (CAD) model [4]. This process is associated with melting the polymer filament using the hotend assembly in a 3D printer and moving the hotend assembly using a multi-axes mechanism that is equipped with stepper motors [2]. The filaments are deposited on a heated bed and cooled using a fan after deposition. The final product can have complex geometries, reduced material waste, a faster fabrication time, and an easier process to be modified. It is known that 3D printing is no longer a prototyping technique since it is being used as a fabrication technique for many industries. However, there are still challenges and unknowns that are inherited by this technology. For example, the material properties of 3D-printed parts are not any more equal to bulk material properties since the bonding strength of the layers dominates the material properties of the actual material

used to print. On the other hand, for advanced fabrication facilities where the qualities of the part (i.e., surface roughness, strength, dimensional tolerances) are critical, the optimum printing conditions are not yet fully defined. In the past decades, the focus of the research field has been attempting to answer these challenges [5–7].

One of the main challenges or limitations of 3D printing is the need for having support materials during prints. For any curvature above 45 degrees, the common practice requires depositing a support material that needs to be removed via mechanical or chemical agitation after the prints. Besides the fact that the support material increases the material waste and fabrication time, it is known that the debonding of support materials causes surface roughness and even damage to the 3D-printed parts. This is more prone to occur if the 3D-printed parts are smaller in dimension (i.e., closer to the resolution limits of the 3D printer). Therefore, we have decided to focus on designing and fabricating special 3D-printed parts that do require support materials to be printed but cannot practically have the supports separated after the prints since they can damage the actual part.

During a TED conference at MIT in 2012 by Skylar Tibbits, the concept of self-assembly at not only nanoscale but also human scale was shown for the first time [8]. During this talk, he presented various 3D-printed parts that could change their morphologies based on the environment they were exposed to. The stimuli presented there were heat, gravitational potential energy shock, and humidity. However, for all the parts, the joints that were meant to change shape were either printed with a different material or printed at a certain geometry. Since then, research in 4D printing has increased exponentially all over the world [4,9–12]. One of the basic characteristics of 4D printing is that it is not static and can reshape with time with the pre-programmed command from the computer or the model designed. This results in a new era of printing involving a new dimension in 3D printing, known as time; hence, 4D printing.

Four-dimensional printing, a relatively new and emerging field, has garnered significant attention in the scientific and engineering communities. It holds promising potential for various applications due to its ability to create dynamic, shape-changing structures that respond to external stimuli. While the field is still evolving, there have been notable advancements and research findings. For example, researchers have focused on developing new materials and fabrication techniques for 4D printing. This includes the use of stimuli-responsive polymers, shape memory materials, and functional inks [4,13]. In these efforts, the materials and properties when exposed to the desired stimuli are studied. Additionally, 4D printing has not been limited to only FDM 3D printing with the time variant added to it. Four-dimensional printing has been explored in inkjet-based, stereolithography, and direct laser writing, each with its own advantages and limitations [10,14]. The main challenge to this day has been the ability to predict the behavior of 4D-printed material. Therefore, there is an extensive effort in the field to develop accurate models and optimize design and simulations to have precision and control over fabricated structures [12].

As discussed by Tibbits in 2013, shape memory polymers (SMPs) exhibit reversible shape changes in response to stimuli like temperature, light, or moisture, enabling complex and programmable 4D structures [8]. SMPs and hydrogels have been widely investigated for 4D printing applications [2,15]. Hydrogels, which can respond to environmental conditions, have shown potential for applications in soft robotics and biomedical engineering [10].

Four-dimensional printing introduces an innovative approach to crafting intricate 3D structures. Unlike conventional methods that directly build 3D geometric features, 4D printing allows for the creation of initially flat precursor structures that can be morphed into the desired 3D shape when triggered by external stimuli. This methodology offers substantial advantages in engineering applications, notably in terms of efficiency and adaptability. Firstly, additive manufacturing can experience significant reductions in build time and process complexity when working with flat-configured printed objects. Moreover, the compact nature of the flat precursor structure facilitates transportation and storage.

The design of precursor structures hinges on active units, pivotal components that govern the ultimate 3D architecture through their out-of-plane deformations. However, generating non-uniform out-of-plane deformations using pure active materials (e.g., hydrogels or liquid crystal elastomers), which inherently prefer homogeneous responses to external stimuli, poses a challenge. To transform the homogeneous deformation of individual materials into non-uniform deformations within the overall structure, the deliberate pre-embedding of internal strain mismatch within active units is necessary, guided by mechanical principles. Thus, the formulation of a mechanical design strategy is crucial for determining the functional attributes of 4D-printed products.

Recent years have witnessed comprehensive reviews of 4D printing advancements, examining various facets such as printing materials [16–19], techniques [16,19,20], and programming pathways [21–23]. For instance, Kuang et al. [17] conducted a material-focused review categorizing existing 4D printing endeavors into those achieved with a single material versus multiple materials. Melly et al. [20] summarized 4D printing methods based on additive manufacturing techniques. However, a thorough exploration of mechanics-driven design strategies guiding the active transformation of 4D-printed structures is noticeably absent from the literature. Bending stands as the predominant method for inducing out-of-plane transformations in 4D printing. In this type of design, the central concern revolves around inducing strain mismatch throughout the thickness of the flat active unit upon exposure to external stimuli. The strain gradient possesses the potential to disrupt the original force equilibrium and drive the active unit into a new equilibrium state characterized by minimized potential energy.

Because of all the above-mentioned strengths and capabilities of 4D printing, it has found applications in various fields including aerospace, architecture, biomedical engineering, and wearable technology [4,19]. In aerospace, 4D-printed structures can adapt to changing environmental conditions, enabling shape-changing wings or morphing structures. In the medical field, 4D-printed implants and scaffolds with controlled drug release capabilities have been explored for personalized medicine and tissue engineering applications [24]. Architectural applications include adaptive structures and responsive facades that can self-assemble or change shape based on environmental factors. Prior research has not compared different 3D printing materials with each other for 4D applications.

In this paper, we focus on a comprehensive experimental effort to understand the behavior of commercially available FDM 3D printing filaments (ABS, PETG, and PLA). Based on the current studies conducted in the field, a comprehensive study that focuses on all three different types of filaments, with different ranges of temperature stimuli, with the analysis of not only radius of curvature but also velocity, is conducted. This paper is novel since it is focused on answering the following questions: (a) Which of the filaments respond more drastically to increased temperature? (b) Does cyclic heating change the material properties of the filaments? (c) Can we develop a model that can predict the radius of curvature of a flat 3D-printed part after exposing it to heat? To answer these questions, a set of samples are 3D-printed using ABS, PETG, and PLA with consistent geometry. Each sample is heated and cooled three times for the three different temperatures of 85 °C, 105 °C, and 125 °C. During the heating and cooling phases, the motion of the 3D-printed parts is captured using a camera. The videos are image-processed by providing quantitative Cartesian coordinates to the end points of the curled parts. For each sample, the displacement and velocity in both x and y coordinates are captured and analyzed. Additionally, the radius of curvature for each sample is captured.

## 2. Materials and Methods

As mentioned above, three different filaments that are commercially available were used for this study: ABS, PETG, and PLA. The printing conditions for each filament are provided in Table 1. These are mostly the printing conditions provided via the Raise3D slicer software (ideaMaker 4.4.3). It should be noted that the inner wall refers to the green central area of Figure 1a, while the outer wall refers to the red strips shown in Figure 1a.

**Table 1.** Printing conditions for each filament type.

| Material | Nozzle Temp (°C) | Bed Temp (°C) | Inner Wall (Green) Speed (mm/s) | Outer Wall (Red) Speed (mm/s) |
|---|---|---|---|---|
| ABS | 250 | 100 | 60 | 40 |
| PETG | 255 | 60 | 60 | 25 |
| PLA | 205 | 60 | 40 | 25 |

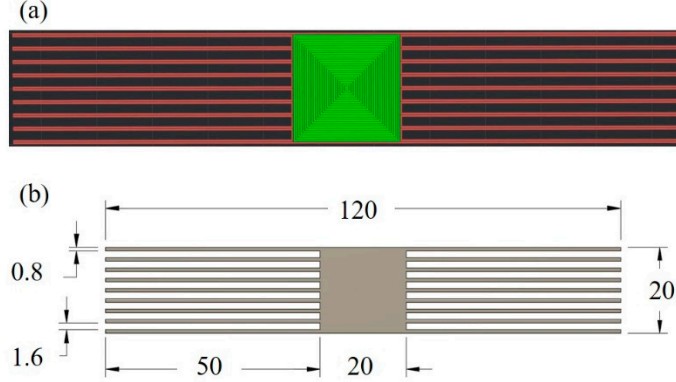

**Figure 1.** Schematic of 3D-printed parts: (**a**) 2D CAD drawing and (**b**) dimensions of the 3D-printed parts are in mm.

For each of the above materials, there were five parts that were 3D-printed using a Raise3D Pro2 printer purchased from Raise 3D Technologies, Inc. (Irvine, CA, USA) with 0.3 mm layer height, 0.4 mm nozzle. For example, after loading the 3D printer with ABS filament, we printed the samples based on the geometry provided in Figure 1 and printing conditions given in the ABS row of Table 1. Each sample was made out of a single type of material. Each configuration was printed 5 times in order to study the consistency of the prints as well. All parts were printed with the green part at the center and outwards with the red strips printed as a continuous line. The green area acted as the base of the parts during the curling process. It should be noted that the samples were heated using the hot plate of the 3D printer while the environments around the samples were room conditions. They were cooled down by turning off the hot plate and letting them stabilize in room conditions.

As mentioned, the samples were printed flat. After the printing process was performed, the samples were moved to the station where the camera was set. Just before the 15 min heating would end, the camera was set to begin recording at 1080 pixels at 30 frames per second (fps). The camera model was Nikon D750 (Tokyo, Japan) equipped with a 105 mm f/2.8D AF Micro-Nikkor lens. The thermal measurements were conducted with a thermal camera FLIR model TG267 (Wilsonville, OR, USA). When the heating period (i.e., 15 min) ended, the sample currently being heated was removed from the heated bed using a plastic scraper and a dental pick. The sample was then placed in a given location (i.e., consistent for all samples). The samples were filmed until approximately one minute before the next sample finished heating. During this time period, it was observed that samples would reach room temperature.

To analyze the videos and track the motion of the samples, Python scripts were developed and used. Each video was then broken into frames, and frames were fed into the code. Each frame was divided into left and right portions and the tip of each sample was tracked through the code during recording time. Each frame would have the x and y coordinates with (0, 0) as the origin. For the calibration of length, each sample (i.e., the left or right side of the actual 3D-printed part) was opened using GIMP 2.10.34 software and the number of pixels referring to 20 mm length was recorded. The csv for each sample was

then opened in another Python script that converted it back into a Pandas data frame. First, any frames where no points were identified and manually entered dead zones of poorly tracked frames were removed from the data. The x and y origin values were subtracted from the x and y coordinates of the tracked point in each frame, and the resulting values were divided by the number of pixels per mm (as determined from GIMP). This resulted in arrays of the x and y positions of the tracked point in mm from the origin. The position arrays then had a moving average of 101 applied to them to smooth the data. The x and y velocity arrays were then found by taking the gradient of the corresponding position array and the time array. The velocity arrays then had another moving average applied to them for graphing. The position and velocity arrays were then able to be graphed against time using the Python library Matplotlib. Additionally, the equation of curvature was solved by using fsolve from the SciPy optimization Python library to determine the value of *r*. Since the optimization would sometimes return a negative solution, the absolute value of the optimized number was used.

The above steps result in arrays of the x and y positions of the tracked point in mm from the origin. The velocity was found by taking the gradient of the corresponding position array and the time array.

To determine the change in radius of curvature over time, the chord length of each frame was first calculated using the Pythagorean theorem on the x and y positions of the tracked point in each frame. Knowing the chord length and the arc length (which was 50 mm based on the printed part), the radius of curvature at each point could be calculated knowing the following relationship in Equation (1), where *r* is the radius, *c* is the chord length, and *a* is the arc length:

$$2r\sin\left(\frac{a}{2r}\right) - c = 0 \tag{1}$$

## 3. Results

In order to clarify the need for 4D printing for the specific geometry that is designed, Figure 2 is put together. If the desired shape has parts that are too thin and require support material, removing support after the print becomes impossible, causing breakage in the actual parts. However, with 4D printing, one can print a part flat, and by fluctuating the temperature, the part can get into the desired curvature.

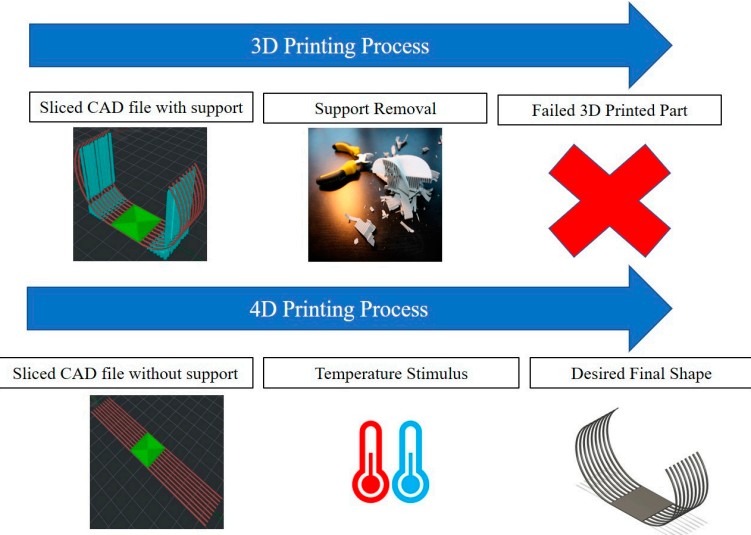

**Figure 2.** Three-dimensional printing versus 4D printing processes.

As mentioned, the samples were printed flat. After the printing process was performed, the samples were moved to the station where the camera was set. Just before the 15 min heating would end, the camera was set to begin recording at 1080 pixels at 30 frames per

second (fps). When the heating period (i.e., 15 min) ended, the sample currently being heated was removed from the heated bed using a plastic scraper and a dental pick. The sample was then placed in a given location (i.e., consistent for all samples). The samples were filmed until approximately one minute before the next sample finished heating. During this time period, it was observed that samples would reach room temperature.

Figure 3 represents the left side of ABS, PETG, and PLA samples for the three different heating temperatures of 85, 105, and 125 °C. This figure is a visual representation of how curling would occur in these samples with the tracking point at the tip of each sample. For each configuration, there were three samples analyzed. Firstly, it is clear that ABS has a repeatable behavior for lower heating temperatures compared to PLA. Additionally, the change in curvature is visually shown to be greater for PLA compared to PETG and ABS. However, in order to have quantitative results, Figure 4 is provided. It should be mentioned that in order to control the undesired curling of the other parts of the 3D-printed part, a thick middle section was printed in our samples. These sections were not attached to the paper; therefore, there was no change in the thermal coefficient of expansion; hence, they acted as the base of the parts.

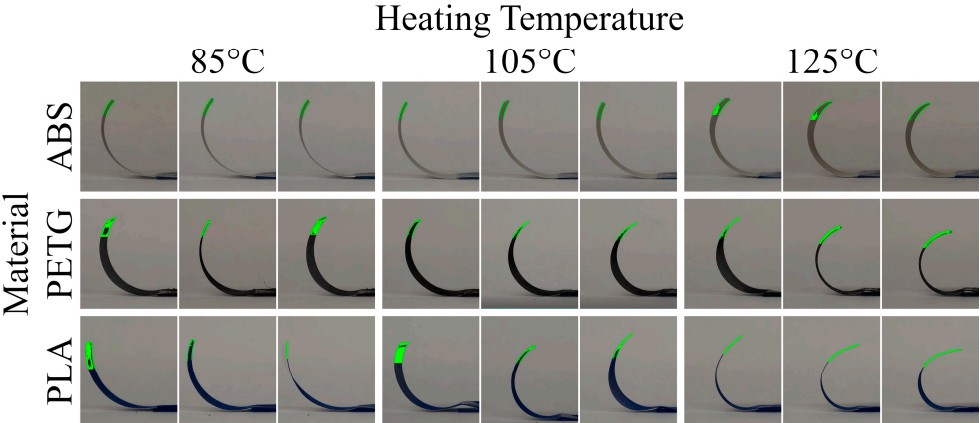

**Figure 3.** Visualization of curling process for ABS, PETG, and PLA samples under three different heating conditions: 85, 105, and 125 °C.

Figure 4 represents the absolute x values for each sample as the samples were cooling down. It should be noted that they all start from the farther end of samples, which is away from the origin (0,0), which is defined to be close to the center of the 3D-printed part, and they travel toward the center. Based on Figure 4's results, the minimum traveled distances are experienced by PLA and ABS for lower heating temperatures, while the maximum traveled distances are observed in PETG and PLA at higher temperatures. This is shown by comparing Figure 4A,G with Figure 4F,I. This is a clear indication that PLA is more responsive to temperature dosage during the curling process compared to ABS and PETG. This is shown by changing temperatures for each sample. The changes in ABS and PETG as we increase the temperature are smaller compared to those in PLA. For PLA samples, the change in displacement from 85 degrees to 105 degrees and finally to 125 degrees is more distinct than the ones shown in ABS and PETG. However, in order to understand how quickly they respond, the velocity graphs are also shown in Figure 5.

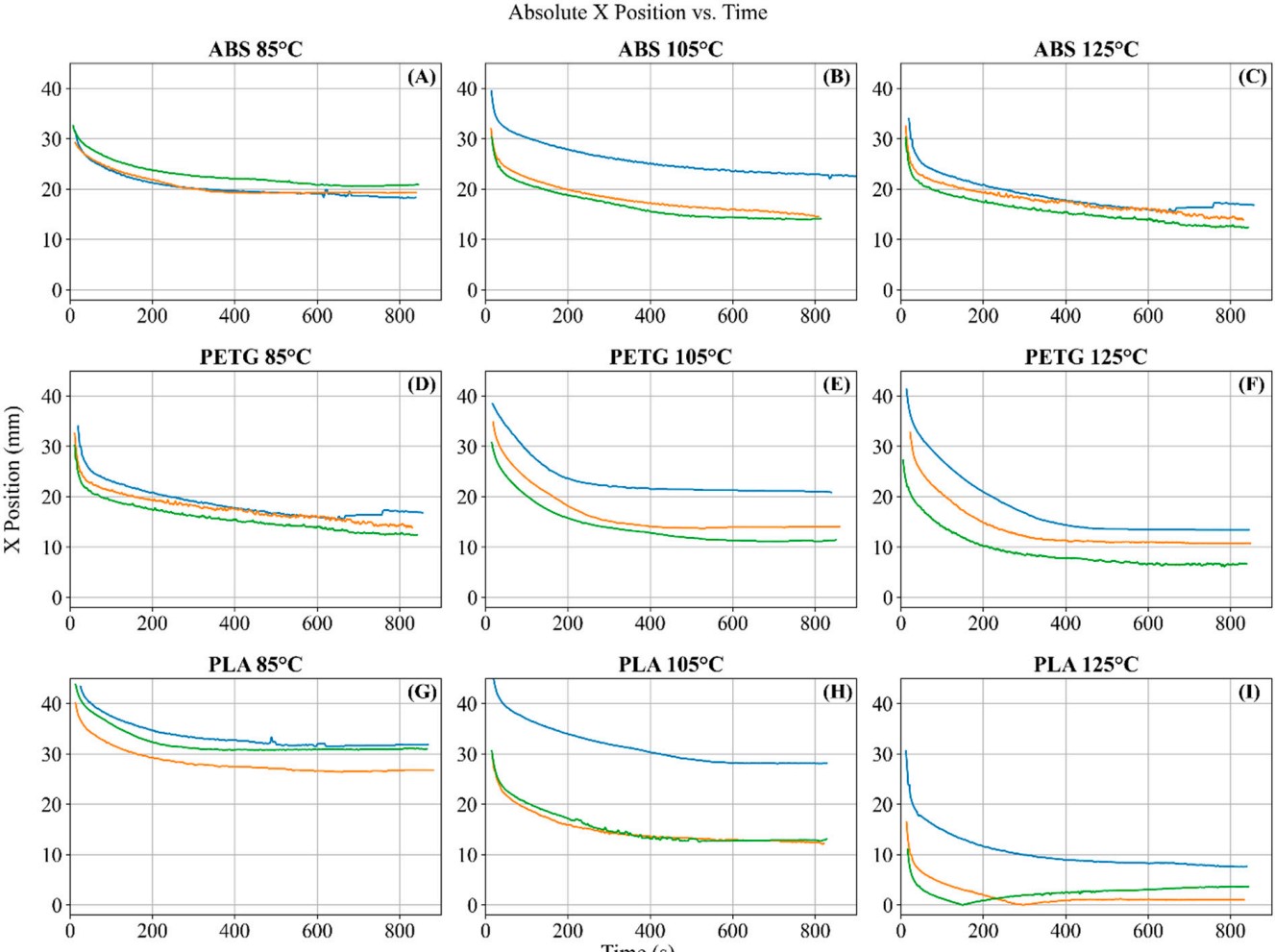

**Figure 4.** Absolute x position vs. time: different colors on each plot represent different samples studied in each category: (**A**–**C**) are ABS material heated at 85 °C, 105 °C, and 125 °C, respectively. (**D**–**F**) are PETG material heated at 85 °C, 105 °C, and 125 °C, respectively. (**G**–**I**) are PLA material heated at 85 °C, 105 °C, and 125 °C, respectively.

Figure 5 shows that as time passes during the cooling process, the rate of change of curling slows down for all of the samples regardless of the initially heated temperature. However, PLA and ABS have higher velocities at higher temperatures during the first 100 s compared to when they are heated to only 85 °C. Therefore, if one needs a faster curling process to occur, they might need to expose the PLA and ABS to higher temperatures. This does not necessarily hold true for PETG samples. Their rate of change does not seem to be a function of initial temperature.

In order to have a full visualization of the motion of the 3D-printed strips, Figure 6 is developed. On each plot, the vertical axis is the y position, the horizontal axis is the x position, and the color gradient represents the time (i.e., starting from light to darker colors as time passes). It is clear that PLA is more responsive to temperature changes and ABS has the minimum response. Additionally, it can be seen that the horizontal motion of the end of 3D-printed parts is more drastic than the vertical axis. This can indeed be due to the geometry defined for the parts.

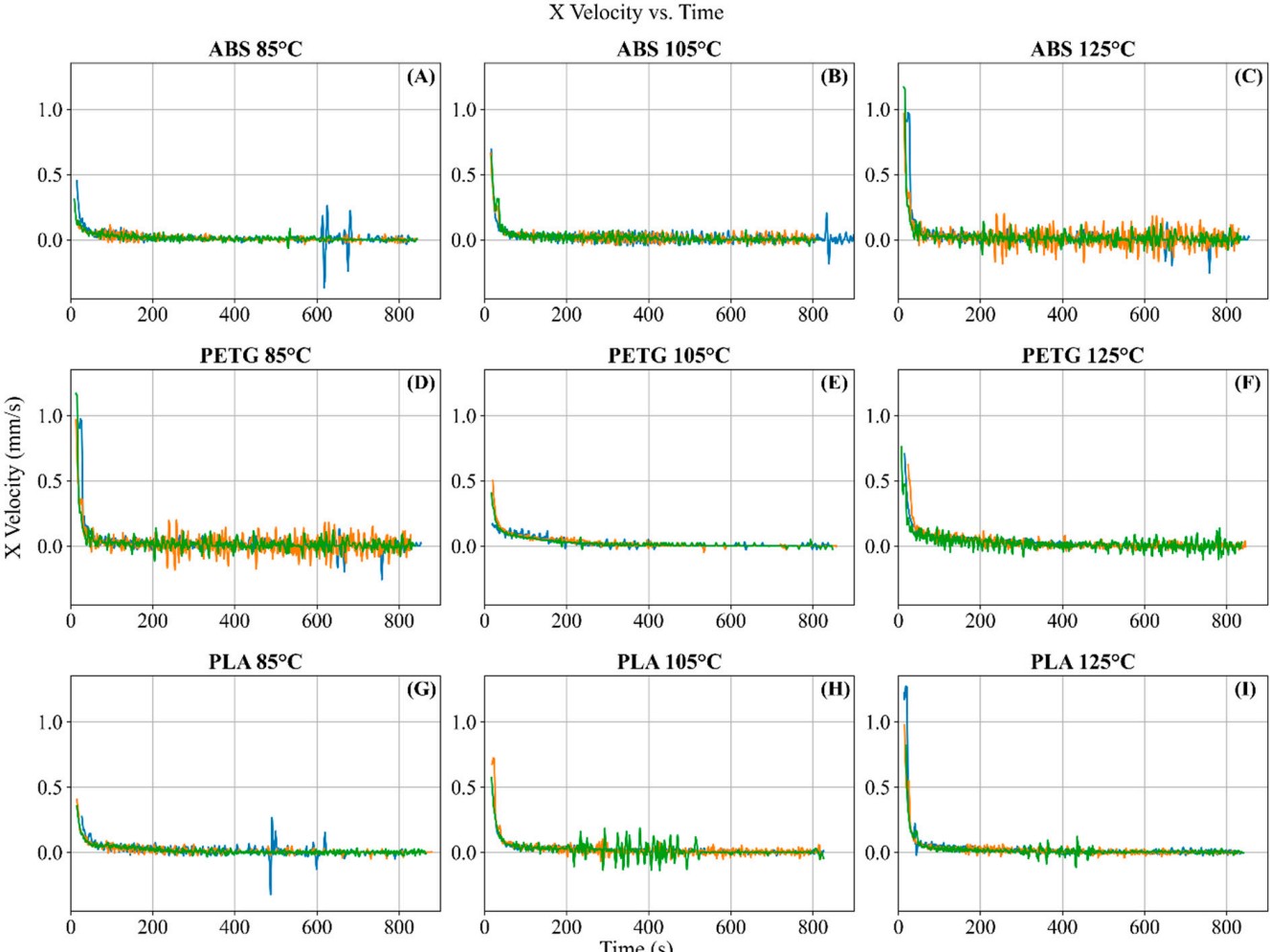

**Figure 5.** Absolute x velocity versus time: different colors on each plot represent different samples studied in each category: (**A**–**C**) are ABS material heated at 85 °C, 105 °C, and 125 °C, respectively. (**D**–**F**) are PETG material heated at 85 °C, 105 °C, and 125 °C, respectively. (**G**–**I**) are PLA material heated at 85 °C, 105 °C, and 125 °C, respectively.

The other parameter that is crucial to the design of circular geometries is the radius of curvature. Figure 7 represents how the radii of curvature for each material (ABS, PETG, and PLA) were changing given different temperature dosages as time passed. It should be emphasized that drastic changes occurred in the first 50 s. However, the parts were still moving (although slowly) until the end of each curve. It is a general trend for all materials that a higher temperature would cause a lower radius of curvature during the cooling process. Additionally, the rate of change of curvature for PLA was much higher than that of other materials.

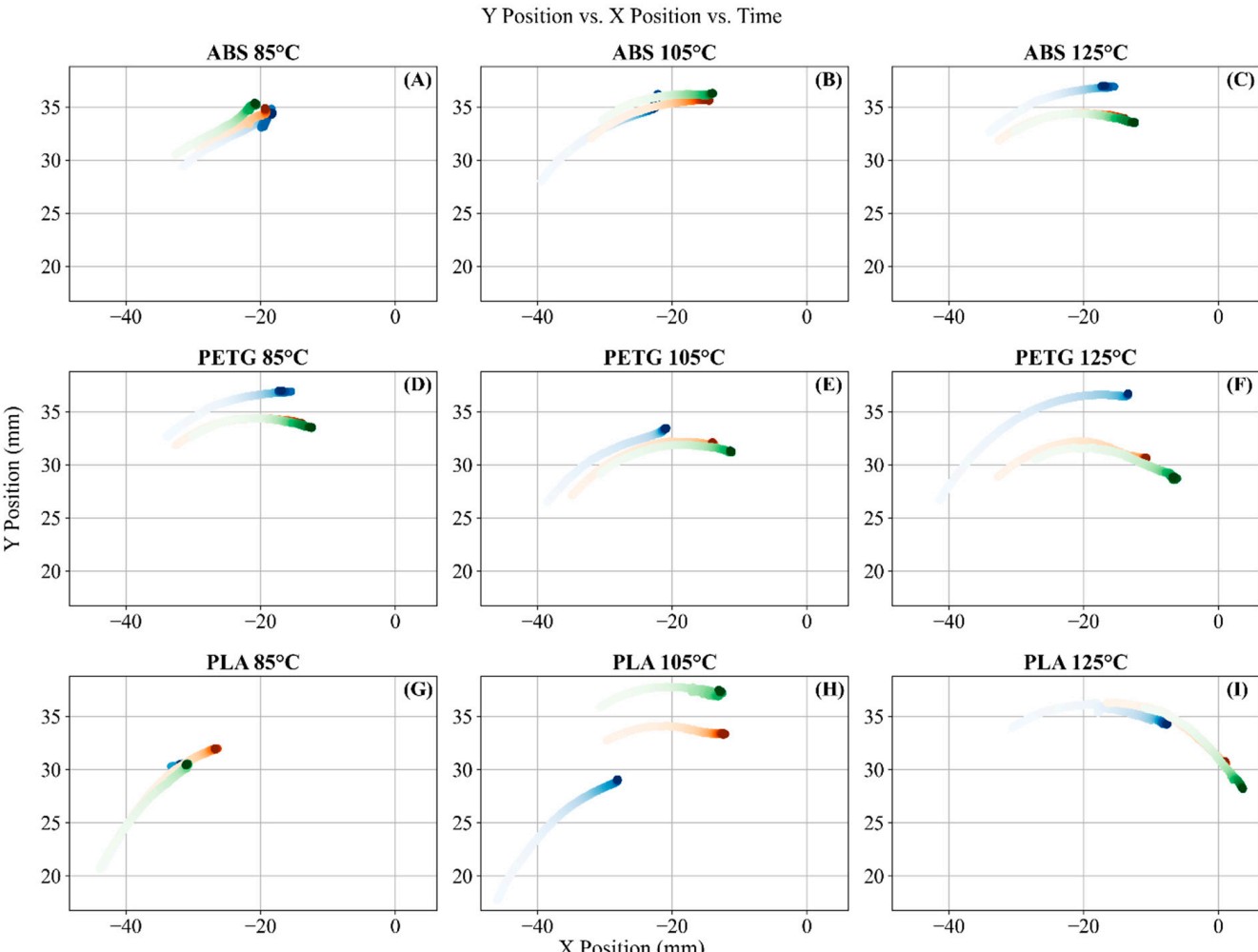

**Figure 6.** Y position versus x position versus time (time is represented by the color gradient going from light to dark): (**A–C**) are ABS material heated at 85 °C, 105 °C, and 125 °C, respectively. (**D–F**) are PETG material heated at 85 °C, 105 °C, and 125 °C, respectively. (**G–I**) are PLA material heated at 85 °C, 105 °C, and 125 °C, respectively.

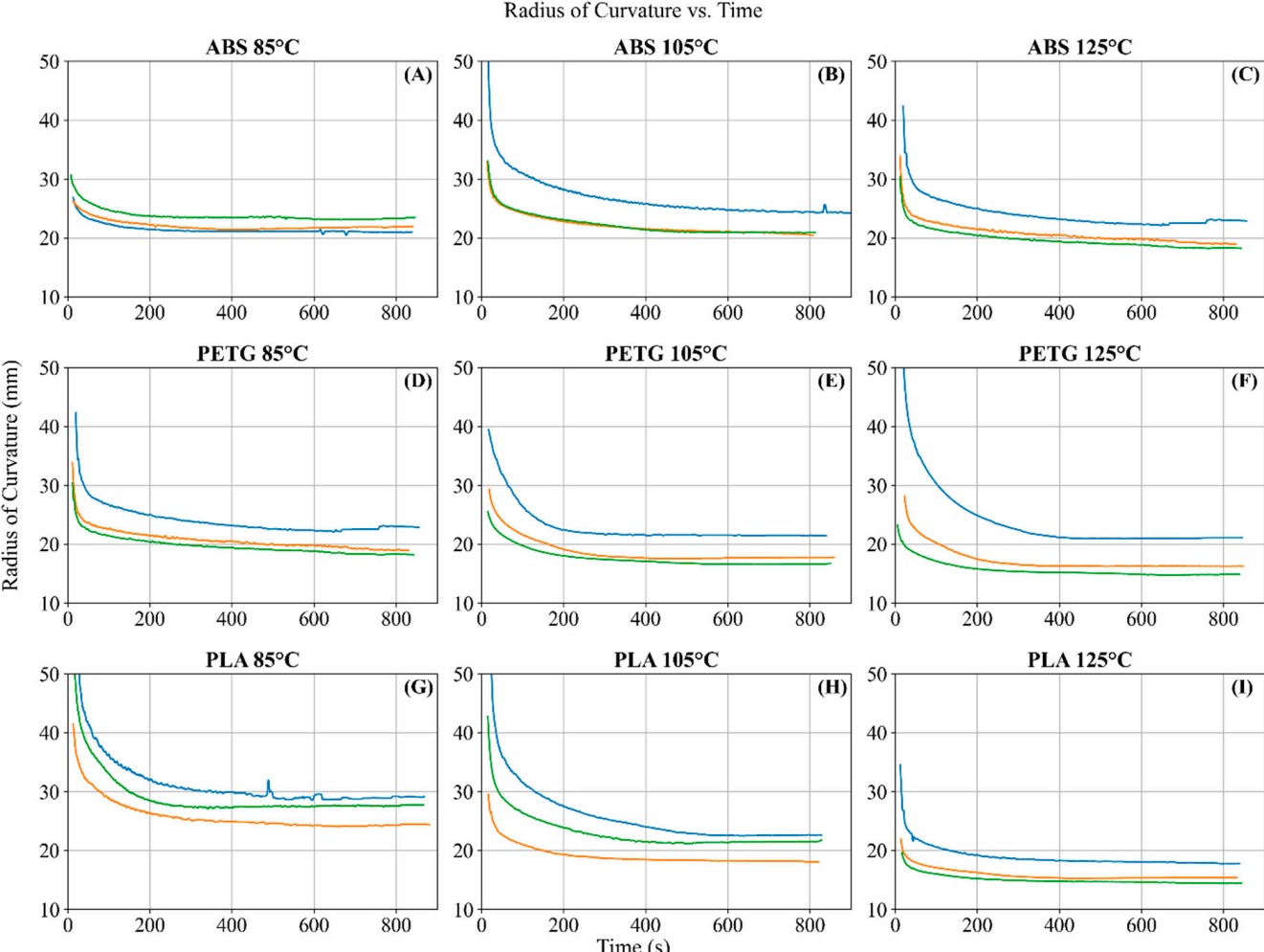

**Figure 7.** Radius of curvature versus time: (**A**–**C**) are ABS material heated at 85 °C, 105 °C, and 125 °C, respectively. (**D**–**F**) are PETG material heated at 85 °C, 105 °C, and 125 °C, respectively. (**G**–**I**) are PLA material heated at 85 °C, 105 °C, and 125 °C, respectively.

Finally, each sample was heated and cooled down three times in order to study the effect of multiple heating and cooling processes on the properties of the polymers. For each configuration, the maximum x position and radius of curvature were averaged and plotted as shown in Figure 8. It is clear that PETG does not necessarily curl more if heated to a lower or higher temperature. On the other hand, PLA is heavily dependent on the heating temperature. Additionally, it is noted that the curling phenomenon gets diminished as we increase the number of heating and cooling cycles. ABS's 3D-printed parts follow the same trend as PLA's but with lower changes in magnitudes. Therefore, if the application requires a fast and drastic response of 3D-printed parts to temperatures, PLA would be ideal. If they want a slow but continuous response, ABS is recommended. PETG can perform well in 4D printing; however, lower or higher temperature does not improve its performance. Additionally, it should be noted that although cycling the heating and cooling procedures might be beneficial for achieving the desired geometry, the strength of the final parts might be diminished. This is studied in more depth by Diaz-Rodriguez et al. [25]. Finally, it should be noted that a higher number of cycles can reduce the maximum range of displacement provided by the samples. For example, PLA samples heated to 85 degrees had around 35 mm of maximum x travel distance, while that parameter got reduced to 30 mm and 26 mm for the second and third heating and cooling cycles, respectively. The materials tested have shown they have the ability to have reversible properties. Therefore,

they are in elastic range and come back to their original shapes after the cycles. However, multiple cycling has reduced their recovery ability.

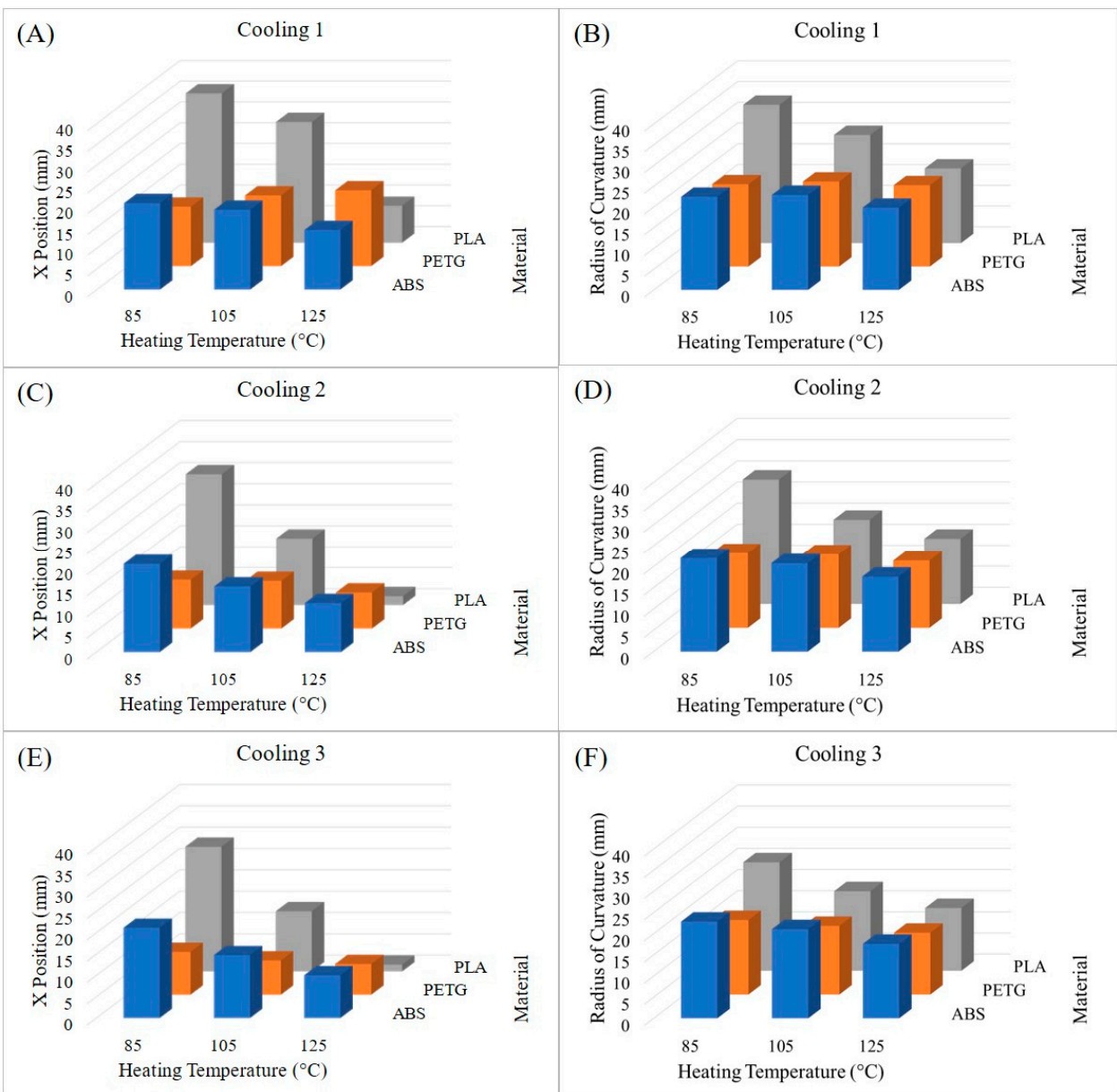

**Figure 8.** Final x position and radius of curvature for ABS, PLA, and PETG materials during three cycles of cooling for three different temperatures: 85, 105, and 125 °C: (**A**,**C**,**E**) are x positions for three materials (ABS, PETG, PLA) for different heating 85, 105, and 125 °C corresponding to first, second and third cycles, respectively. (**B**,**D**,**F**) are radius of curvature for three materials (ABS, PETG, PLA) for different heating 85, 105, and 125 °C corresponding to first, second and third cycles, respectively.

## 4. Conclusions

This paper has focused on investigating the response of common FDM 3D-printed polymers (i.e., ABS, PETG, and PLA) to temperature after the parts are fabricated. Each designed sample was heated and cooled to 85, 105, and 125 °C and room temperature, respectively. The complete motion of each sample was recorded in x and y coordinates and led us to measure the instanton radius of curvature for all samples. It was found that PLA is the sample most sensitive to the temperature stimulus after 3D printing is performed. The curling increased as the heating temperature increased. ABS showed minimum response to

curling. These findings can be useful when deciding which material to choose that will go through heating cycles during application. For example, if the design requires minimum expansion or change in shape, ABS might be a better choice compared to PLA or PETG. However, if one requires having the part form a specific shape that is not possible with 3D printing but doable with 4D printing, PLA seems to be the better choice. It should also be noted that although these trends hold true, the quantitative values are dependent on the design of the 3D-printed parts presented in this paper. This paper shows 4D printing is a promising technology that can have different applications. For example, the self-assembly of 3D-printed parts when exposed to the desired stimulus can drastically reduce fabrication and assembly times. For example, soft robotics can benefit from automated hinging and locking mechanisms, packaging, and assembly processes. Four-dimensional-printed parts can be useful for space applications as well. For example, 3D-printed flat parts can be printed on earth and be transformed into their final shape in space.

**Author Contributions:** Conceptualization, R.M. and B.E.; methodology, R.M.; software, R.M.; validation, R.M. and B.E.; formal analysis, B.E.; investigation, B.E.; resources, B.E.; data curation, B.E.; writing—original draft preparation, R.M. and B.E.; writing—review and editing, B.E.; visualization, R.M.; supervision, B.E.; project administration.; funding acquisition, B.E. All authors have read and agreed to the published version of the manuscript.

**Funding:** This research was funded by Pennsylvania Department of Community and Economic Development under PA Manufacturing Innovation Program with grant number 1060170-458516.

**Institutional Review Board Statement:** Not applicable.

**Informed Consent Statement:** Not applicable.

**Data Availability Statement:** Data can be available upon request.

**Conflicts of Interest:** The authors declare no conflict of interest.

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
