# Peer review of "Experimental Investigation on Effect of Temperature on FDM 3D Printing Polymers: ABS, PETG, and PLA"

_applsci, doi:10.3390/app132011503_

Round 1
Reviewer 1 Report
The manuscript tests the final shape and 3D-printed polymers after exposure to different temperatures. The intended use of the proposed process is for 4D printing parts for which a shape change is quantified in terms of temperature exposure. The paper is well-written, and the literature is comprehensive. Some details need to be addressed before it might be considered for publication.
Minor changes
The authors mention 4D printing, where changes in shape are sought, but the way the warping is presented in the literature review seems undesirable. It lacks a transition in using the unwanted distortion for shape change.
The paragraph between L 194 and L202 is repeated.
Please give details about the camera´s lens optics and the instrument used to measure temperature.
The samples of Fig 3 show the cartesian tip coordinate as a measure of curling. Figs 4 and 5 show the X and Y coordinates over time for different materials and heating temperatures. Are they not related variables? If so, Figs 4 and 5 are redundant.
Perhaps Fig 6 can be improved by adding labels for each heating temperature to see the evolution of tip position with heating temperature.
Major issues
While the literature review is comprehensive, I think the paper of Pei and Loh, 2018 that discusses the advantages and limitations of 4D printing should be considered.
Although the proposed process parameters might help the assembly and find the final shape of a 4D part, one unintended consequence of such a process might be the modification of mechanical properties (see Diaz, 2023). I believe it is an outcome that readers should be aware of.
Pei, E., Loh, G.H. Technological considerations for 4D printing: an overview. Prog Addit Manuf 3, 95–107 (2018). https://doi.org/10.1007/s40964-018-0047-1
Diaz, Pertuz, Bohorquez. Impact resistance for 3D printed polymer composites under temperature changes. Journal of Manufacturing and Materials Processing. (2023), 7, 178. https://doi.org/10.3390/jmmp7050178
Author Response
We thank reviewer's valuable comments. We have implemented the changes and are attaching the point-by-point response here.

Reviewer 2 Report

Moderate editing of the English language required
Author Response

(The authors gave the same response as above.)

Reviewer 3 Report
Dear Authors,
You present an interesting topic in your article. However, some aspects require further clarification and description to strengthen the manuscript:
- Lines 154-158: Please expand on the printing process description:
- For each sample, was one material used or were multiple materials combined?
- If multiple materials were used, was there any break time between material changes?
- Please explain the principle behind the reshape ability of your model in more detail.
- Figure 1 needs more description of the model parts. Does the initial shape of the model impact the shaping ability in terms of geometrical changes during shaping and reshaping?
- Figure 2: How can you ensure the model deforms as shown in the last picture? Is there a relationship between the shape geometry and the 4D ability (shaping and reshaping)?
- Figure 3 shows testing of the shaping ability at different temperatures. Please describe the environment and heating process in more detail.
- Your results show the part can be shaped as in Fig. 3. How about the reshaping ability - what is the shape of the sample when it returns to room temperature?
- Overall, the discussion of results is still limited. Please expand on your results and discuss in depth the potential positive applications enabled by your findings.
I hope these suggestions help strengthen the manuscript. Please let me know if you would like any clarification or have additional questions.
Sincerely,
Author Response

(The authors gave the same response as above.)

Round 2
Reviewer 1 Report
The authors have improved the paper. I think it has sufficient novelty and is attractive to the readers. I recommend its publication on Applied Sciences with these minor changes.
L313. "The temperatures selected for heating and cooling do not cause any plastic and permanent changes on samples". That statement cannot be concluded from the performed tests.
L 238. samples for sampled
L 239. add THE thermal coeffcient ...
Author Response
We appreciate reviewer's comments. We have done the following changes:
(1) Deleted the sentence in L313 that was mentioned.
(2) Fixed "Samples" in L238.
(3) Added "the" in L239.
Reviewer 2 Report
Accept after minor text editing.
Accept after minor text editing.
Author Response
We appreciate reviewer's comments. We have proofread our manuscript and fixed any grammar or spelling issues we could find.
Reviewer 3 Report
Dear Authors,
Thank you for your reply and modifications to the latest version of the manuscript. Overall, the paper is suitable for publication now. However, I still have some remaining concerns:
The novelty of your research lies in using the 3 commercial materials (specify materials here). Please clarify in the introduction how there is limited prior research on these specific materials for this application.
The title mentions 4D printing, which implies the printed part can reshape itself over time. However, the current scope focuses more on the shaping with different materials. Consider revising the title to more accurately reflect the key contributions around multi-material 3D printing.
Please address these final concerns before proceeding to publication. I believe doing so will strengthen the manuscript.
Sincerely,
Author Response
We appreciate the reviewer's comments. We have addressed them as following:
(1) We added a sentence in introduction in Line 134 that reads: "Prior research...".
(2) We changed the title of manuscript and removed "4D Printing" and added the different types of polymers.